# Assessment of Fallopian Tube Epithelium Features Derived from Induced Pluripotent Stem Cells of Both Fallopian Tube and Skin Origins

**DOI:** 10.3390/cells12222635

**Published:** 2023-11-16

**Authors:** Yu-Hsun Chang, Kun-Chi Wu, Kai-Hung Wang, Dah-Ching Ding

**Affiliations:** 1Department of Pediatrics, Hualien Tzu Chi Hospital, Buddhist Tzu Chi Medical Foundation, Tzu Chi University, Hualien 97004, Taiwan; cyh0515@gmail.com; 2Department of Orthopedics, Hualien Tzu Chi Hospital, Buddhist Tzu Chi Medical Foundation, Tzu Chi University, Hualien 97004, Taiwan; drwukunchi@yahoo.com.tw; 3Department of Medical Research, Hualien Tzu Chi Hospital, Buddhist Tzu Chi Medical Foundation, Tzu Chi University, Hualien 97004, Taiwan; kennyhug0201@gmail.com; 4Department of Obstetrics and Gynecology, Hualien Tzu Chi Hospital, Buddhist Tzu Chi Medical Foundation, and Tzu Chi University, Hualien 97004, Taiwan; 5Institute of Medical Sciences, Tzu Chi University, Hualien 97004, Taiwan

**Keywords:** induced pluripotent stem cell, fallopian tube epithelium, ovarian cancer, stem cell, high-grade serous cancer

## Abstract

Fallopian tube epithelial cells (FTECs) play a significant role in the development of high-grade serous ovarian cancer (HGSOC), but their utilization in in vitro experiments presents challenges. To address these limitations, induced pluripotent stem cells (iPSCs) have been employed as a potential solution, driven by the hypothesis that orthologous iPSCs may offer superior differentiation capabilities compared with their non-orthologous counterparts. Our objective was to generate iPSCs from FTECs, referred to as FTEC-iPSCs, and compare their differentiation potential with iPSCs derived from skin keratinocytes (NHEK). By introducing a four-factor Sendai virus transduction system, we successfully derived iPSCs from FTECs. To assess the differentiation capacity of iPSCs, we utilized embryoid body formation, revealing positive immunohistochemical staining for markers representing the three germ layers. In vivo tumorigenesis evaluation further validated the pluripotency of iPSCs, as evidenced by the formation of tumors in immunodeficient mice, with histological analysis confirming the presence of tissues from all three germ layers. Quantitative polymerase chain reaction (qPCR) analysis illuminated a sequential shift in gene expression, encompassing pluripotent, mesodermal, and intermediate mesoderm-related genes, during the iPSC differentiation process into FTECs. Notably, the introduction of WNT3A following intermediate mesoderm differentiation steered the cells toward a FTEC phenotype, supported by the expression of FTEC-related markers and the formation of tubule-like structures. In specific culture conditions, the expression of FTEC-related genes was comparable in FTECs derived from FTEC-iPSCs compared with those derived from NHEK-iPSCs. To conclude, our study successfully generated iPSCs from FTECs, demonstrating their capacity for FTEC differentiation. Furthermore, iPSCs originating from orthologous cell sources exhibited comparable differentiation capabilities. These findings hold promise for using iPSCs in modeling and investigating diseases associated with these specific cell types.

## 1. Introduction

Epithelial ovarian cancer, specifically high-grade serous ovarian cancer (HGSOC), is the fifth leading cause of cancer-related deaths in women worldwide, with a global annual diagnosis of over 220,000 cases [1,2]. Most patients with ovarian cancer have late-stage disease, with a five-year survival rate of only 30–40% [1]. Unfortunately, most ovarian cancer patients are diagnosed at an advanced stage, bearing a five-year survival rate of merely 30–40%. Notably, HGSOC has been found to originate from the fallopian tube epithelium (FTE) [3], but the scarcity of an appropriate research model due to the challenges in retrieving and sustaining FTE in vitro has posed a significant hurdle.

FTE encompasses two crucial types of epithelial cells: secretory and ciliated. The current research models for fallopian tubes, including ex vivo and organoid formation models, strive to maintain cell polarity by reestablishing ciliated and secretory cells [4,5,6]. However, these models face limitations due to the absence of luminal architecture and a conducive microenvironment, leading to restricted fallopian tube epithelial cell (FTEC) proliferation and increased senescence [4,5,6]. An alternative involves employing FTE research in animal models, such as genetically engineered mice and patient-derived xenografts, yet these models may not fully replicate the complex evolution of human tumors [7,8,9,10,11].

Furthermore, obtaining FTE requires surgical intervention, making it challenging to retrieve repeatedly [12]. Thus, there is a compelling need for a consistent FTEC line, potentially offering an enduring resource for cancer research. While current FTEC lines primarily focus on cancer initiation, they may not precisely mirror actual genetic changes, emphasizing the necessity for fresh FTECs as a renewable source for research.

Interestingly, FTECs demonstrate stemness, self-renewal capabilities, and Wnt-related organoid formation [13]. They display a cuboidal cell morphology and consistent proliferation up to nine passages [13]. Nevertheless, they tend to become senescent over time. Therefore, genetically modified cell lines have been developed as an alternative [14]. However, the application of fresh FTECs necessitates further exploration.

Recently, researchers have made strides in differentiating FTECs from induced pluripotent stem cells (iPSCs) derived from fibroblasts, offering potential advantages due to the pluripotency, self-renewal, and limitless proliferation of iPSCs [15]. This flexibility has led to successfully modeling various inherited human diseases, including familial cancers [16,17]. It is essential to note that iPSCs derived from different cellular origins may exhibit varying differentiation capacities, underlining the significance of considering the cellular source of iPSCs during clinical translation [18,19].

With the profound potential of iPSCs, this study aimed to develop a new iPSC line originating from FTECs and to compare its differentiation capabilities with those of NHEK-iPSCs. This research seeks to provide a robust foundation for future investigations in the field.

## 2. Materials and Methods

### 2.1. Establishing iPSCs from FTECs

#### 2.1.1. Ethics

The experimental protocol was approved by the Research Ethics Committee of Hualien Tzu Chi Hospital (IRB110-224-C). NHEK-iPSC was already derived in the previous study [20].

#### 2.1.2. FTEC Culture

The culture medium for FTECs consisted of Dulbecco’s Modified Eagle Medium (DMEM, cat. No. D5030, Sigma, St. Louis, MO, USA) supplemented with 10% fetal bovine serum (FBS, cat. No. F7524, Sigma) and 5 μg/mL insulin (Sigma) [13]. The FTECs were then cultured in uncoated culture dishes. The medium was changed every three days. The cells were passaged at a 1:3 ratio when they reached 90% confluence after seven days of culture. Passages 4–6 of the FTECs were used for iPSC induction.

#### 2.1.3. Induction Protocol

iPSCs were generated using the Sendai virus reprogramming system [20]. FTECs were maintained in a culture medium at 37 °C in a 5% CO_2_ incubator. The medium was changed every three days until the cells reached 75% confluence. A four-gene non-integrated introduction system (CytoTune™-iPS 2.0 Sendai reprogramming kit [Carlsbad, CA, USA cat. No. A16517]) was used in the experiments. The system comprised polycistronic *KLF4–OCT3*/*4–SOX2*, *CMYC*, and *KLF4* in the Sendai virus, which delivered these genes into cells. FTECs were cultured at a density of 5 × 10^4^ cells/well (diameter = 34.8 mm). After reaching 40% confluence in 2–3 days, Sendai virus vectors at a multiplicity of infection (MOI) of 4:4:2 (*KLF4–OCT3/4–SOX2: CMYC: KLF4*) were introduced into the medium. After 24 h, the cells were washed with 1X Dulbecco’s phosphate-buffered saline (PBS) (-Ca^2+^/Mg^2+^) (cat. No. 10010023, Gibco, Waltham, MA, USA) and placed in a fresh FTE culture medium. The resulting cells were passaged onto vitronectin (rhVTN-N, cat. No. A14700, Gibco, MA, USA)-coated 6-well plates at a density of 2 × 10^4^ cells/well, seven days post-transfection. The cells were cultured in a chemically defined Essential 8 medium (cat. No. A1517001, Gibco) and 10 μM Y-27632 (cat. No. SCM075, Merck Millipore, Burlington, MA, USA). Twelve days post-transduction, TRA-1-60 and TRA-1-81 colonies were selected for live staining. The selected cells represented first-generation iPSCs. iPSCs were then passaged by manual weekly cutting and cultured on a growth factor-reduced Matrigel Matrix (cat. No. 356234, BD Biosciences, Frank Lakes, NJ, USA)-coated plate at 37 °C in a 5% CO_2_ incubator.

#### 2.1.4. iPSC Pluripotency Characterization

The iPSC was used for the study after passage 5 with the virus vector was eliminated spontaneously (RT-PCR would check the virus vector and related pluripotency genes). iPSC pluripotency was evaluated by immunostaining for the pluripotency markers SOX2 (BioLegend, 65108), TRA-1-60 (Merck Millipore, MAB4360), and TRA-1-81 (Merk Millipore 90233). qPCR was then used to evaluate the pluripotency markers (*OCT4*, *SOX2*, *NANOG*, *KLF4*, and *LIN28*). The iPSC differentiation ability was evaluated using in vitro three germ layer embryoid body differentiation and in vivo teratoma formation. The markers of the three germ layers, brachyury (mesoderm, Abcam ab-20680, Cambridge, UK), Tuj-1 (beta-III tubulin, Gene Tex GTX 631836, Irvine, CA, USA), MAP2 (ectoderm, Gene Tex GTX11267), and ATBF1 (endoderm, Santa Cruz SC-48807, Dallas, TX, USA), were evaluated by immunostaining (1:200).

### 2.2. qPCR

qPCR was performed to confirm the expression of the pluripotency genes. Table 1 lists the primer sequence of the tested genes. Real-time qPCR (RT-qPCR) was performed and monitored using a FastStart Universal SYBR Green Master Mix (cat. No. 03003230001, Roche, Indianapolis, IN, USA) and a qPCR detection system (ABI StepOnePlus system, Applied Biosystems, Foster City, CA, USA). Glyceraldehyde 3-phosphate dehydrogenase (*GAPDH*) was used as an internal control. Each target gene’s expression level was calculated using the 2^−ΔΔCt^ method [21]. Three readings were obtained for each experimental sample and each gene of interest. All of the experiments were performed in triplicate.

The iPSC genes tested were *OCT4*, *SOX2*, *NANOG*, *KLF4*, and *LIN28* (pluripotency markers). *MIXL1, BRACHYURY* (mesoderm markers), *PAX2, OSR1*, and *GATA3* (intermediate mesoderm markers) were determined on differentiation days 0, 2, 4, and 6. *SIX2*, *FOXD1* (kidney markers), *WT1*, and *OVGP1* (MD markers) were tested on differentiation days 0, 2, 4, and 8. Pluripotency genes, including *OCT4*, *SOX2*, and *NANOG*, were tested on differentiation days 0, 2, 4, and 6. RT-PCR checked the Sendai virus vector and related pluripotency genes. The primer sequence of the genes is listed in Table 1.

### 2.3. iPSC IHC and Three Germ Layer Differentiation

iPSCs colonies were cultured on Matrigel-coated chamber slides (cat. No. Z734853, Nunc, Thermo Fisher Scientific, Waltham, MA, USA) and subjected to IHC 3–7 days after passage. The cells were fixed with 4% paraformaldehyde; permeabilized with 0.1% Triton X-100; blocked with 4% normal goat serum; and treated with primary antibodies against OCT4, SOX2, and TRA-1-81 (cat. No. SCR001, ES Cell Characterization Kit; Chemicon, EMD Millipore, Billerica, MA, USA).

For iPSC differentiation, EB formation was performed for five days. The resulting EBs were plated onto gelatin-treated chamber slides and fixed. Antibodies specific for the three germ layers, namely the ectoderm (microtubule associated protein 2 (MAP2) and beta-III tubulin), mesoderm (brachyury), and endoderm (ATBF1), were identified (1:200).

### 2.4. Chromosomal Analysis

Karyotyping of iPS-FTEC cells was carried out at the Cytogenetics Laboratory of Ko’s Obstetrics and Gynecology Clinic (Taipei, Taiwan). In brief, the cells were cultured until they reached exponential growth and were then treated with colchicine (cat. No. C9754, Sigma) to arrest them at metaphase. Subsequently, the cells were subjected to a hypotonic solution to induce bursting. Following bursting, the cells were affixed to a glass slide and stained with Giemsa stain. A cytogeneticist reviewed the chromosomes, which were organized into karyograms. The distribution of chromosome numbers was determined by examining 50 metaphases. The karyotypes of these 50 metaphases were analyzed, and the results were reported following the guidelines of the 2016 International System for Human Cytogenetic Nomenclature.

### 2.5. Animal Teratoma Formation Experiments

All of the animal experiments were performed according to protocols approved by the Institutional Animal Care and Use Committee of Hualien Tzu Chi Hospital. Animal Research: Reporting of In Vivo Experiments (ARRIVE) guidelines and regulations were followed.

Three 6–8-week-old female NOD-SCID mice were purchased from Tzu Chi University. The mice were fed in a temperature-controlled environment (Animal Center at Tzu Chi University) under a 12 h light/12 h dark cycle. Professional personnel at the Animal Center were responsible for animal care.

For the teratoma generation assay, iPSCs were removed from the dish by mechanical slicing using glass capillaries, pelletized, and resuspended in PBS.

iPSCs (5 × 10^5^) mixed with Matrigel (1:1) were injected into the subcutaneous tissue of the mice. Tumor formation was followed by palpation. After six months, or after reaching a tumor size of 0.5 cm^3^ in volume, the mice were euthanized with CO_2_ inhalation. After euthanization, the tumors were dissected, fixed, embedded in paraffin, and processed for histological examination.

### 2.6. Histological Examination

The resultant teratomas were fixed in 10% formalin and embedded in 4% paraformaldehyde. The specimens were sectioned into 4 μm thick slices. The tissue sections were stained with hematoxylin and eosin (H&E) (Sigma, St. Louis, MO, USA) to evaluate the cellular architecture. The structures of the germ layers were examined and photographed using a light microscope (Nikon, Tokyo, Japan).

### 2.7. iPSC Differentiation to FTE

The differentiation protocol was followed according to a previous study, with slight modification [15]. The newly developed FTE-iPSCs and previously established NHEK-iPSCs were used for the FTE differentiation experiments (Figure 1).

### 2.8. iPSCs to Mesoderm Differentiation (Day 0–2)

The differentiation medium consisted of DMEM/F12 (11320033, Gibco) + GlutaMax (35050061, Invitrogen, Waltham, MA, USA), 2% FBS supplemented with 10 μM ROCK inhibitor Y-27632 (Stemgent, Beltsville, MD, USA), 100 ng/mL human recombinant activin A (03-0001, Stemgent), 3 μM CHIR99021 (13122, Cayman Chemicals, Ann Arbor, MI, USA), and 500 U/mL penicillin/streptomycin (15140122, Gibco).

### 2.9. Mesoderm to Intermediate Mesoderm Differentiation (Days 2–4)

The cells were cultured in Matrigel-coated dishes. The differentiation medium consisted of DMEM/F12 (Gibco) + Glutamax (Invitrogen) supplemented with 10% Knockout Serum Replacement (10828028, Invitrogen), 0.1 mM non-essential amino acids (11140050, Invitrogen), 0.55 mM 2-mercaptoethanol, 100 ng/mL BMP4 (314-BP, R&D Systems, Minneapolis, MN, USA), 3 μM CHIR99021 (Cayman Chemicals), 10 μM ROCK inhibitor Y-27632 (Stemgent), and 500 U/mL penicillin/streptomycin (Gibco).

### 2.10. Spheroid to Mullerian Duct Epithelium Differentiation (Days 4–6)

The spheroids were selected and plated on Matrigel-coated dishes. Fallopian tube media (FTM) consisting of DMEM/F12 (Gibco), 2% reconstituted Ultroser G (15950-017, Pall), 10 μM ROCK inhibitor Y-27632 (Stemgent), and 500 U/mL penicillin/streptomycin (Gibco) were used.

For FTE differentiation, 100 ng/mL human recombinant WNT4 (6076-WN, R&D Systems) with or without 3 μM CHIR99021(Cayman Chemicals), and 100 ng/mL human recombinant WNT3A (5036-WN, R&D Systems) with or without 3 μM CHIR99021 (Cayman Chemicals) were added.

### 2.11. FTE Differentiation (Days 6–8)

The FTE medium was adjusted by adding human recombinant Follistatin (120-13, Peprotech) (20 ng/mL), estrogen (E2257, Sigma, 1 ng/mL), and progesterone (P8783, Sigma, 33 ng/mL).

### 2.12. Growing FTE Organoids from Spheroids in the Matrigel

On day 4, spheroid growth in the Matrigel was collected using a 200 μL pipette tip and pooled into a 1.5 mL tube. The collected spheroids were mixed with 50 μL Matrigel (BD Biosciences, Franklin Lakers, NJ, USA), estrogen (1 ng/mL), and progesterone (33 ng/mL), and slowly placed into the well of a 24-well dish. The droplets were allowed to solidify for 10–15 min in an incubator at 37 °C. FTM supplemented with growth factors was then added. The medium was changed every 3–4 days. The cells were replated every two weeks.

### 2.13. Organoid IHC

The staining procedure was the same as described in the previous section. The fixed organoids were then embedded in an optimal cutting temperature (OCT) compound (25608-930, Tissue-Tek, Torrance, CA, USA). Cryostat was used on frozen sections (thickness = 12 μm), placed on a glass slide, and stored at −80 °C. The sections were rehydrated with 1X PBS for 5 min and blocked with PBS containing 10% FBS and 0.05% Triton X-100 (PBST) for 1 h at 25 °C, followed by 2 h of incubation with primary antibodies in a blocking solution at 25 °C. The slides were washed thrice with PBS-T for 15 min at a room temperature of 25 °C and incubated with specific AF488- or AF594-conjugated secondary antibodies, followed by 4′,6-diamidino-2-phenylindole (DAPI) counterstaining. After washing thrice with PBST, the slides were covered with a cover slide and imaged using a light microscope. The primary antibody concentration was 1:200, and that of the secondary antibodies was 1:400. The primary antibodies used were PAX8 (CL48860145, Proteintech, Rosemont, IL, USA), FOXJ1 (14-9964-82, Invitrogen), LGR5 (373804, BioLegend, San Diego, CA, USA), and DAPI (D1306, Invitrogen). The complete differentiation protocol is illustrated in Figure 1.

### 2.14. Statistical Analysis

All of the analyses were performed using SPSS Statistics, version 25 (IBM, Armonk, NY, USA). Gene expression at different stages of differentiation was compared in both cell lines. Data are presented as mean ± standard deviation. Means were compared using a one-way analysis of variance followed by Tukey’s correction. The level of statistical significance was set at *p* < 0.05.

## 3. Results

### 3.1. FTEC-iPSCs and Xenograft Tumor Characteristics

FTEC-iPSC could form a colony on the Matrigel (Figure 2A). Immunohistochemistry (IHC) and quantitative polymerase chain reaction (qPCR) were used to demonstrate the pluripotency of FTEC-iPSCs. IHC revealed positive staining for SOX2, TRA-1-60, and TRA-1-81 (Figure 2B). qPCR showed an increased expression of the pluripotency-related genes *OCT4*, *SOX2*, *NANOG*, *KLF4*, and *LIN28* (Figure 2C). RT-PCR showed no virus-related pluripotency gene existed in FTEC-iPSC after passage 5 (P6, Figure 2D). The differentiation capabilities of iPSCs were evaluated based on embryoid body formation. Accordingly, qPCR revealed increased ectoderm markers of *TUBB3* and *MAP2* expression (Figure 2E). IHC revealed positive staining for three germ layer markers: brachyury (mesoderm), ATBF1 (endoderm), and beta-III tubulin and MAP2 (ectoderm) (Figure 2F). The chromosome analysis of iPSC showed normal female karyotypes (Figure 2G). Tumorigenesis was evaluated to prove the pluripotency of iPSCs in vivo. Tumors were formed after three months in nonobese diabetic/severe combined immunodeficiency (NOD-SCID) mice (Figure 2H). The teratoma histology showed three germ layer tissues: endoderm (cartilage and fat tissue), endoderm, and ectoderm (Figure 2I,J).

### 3.2. Sequential Differentiation of FTEC-iPSCs from iPSCs to Intermediate Mesoderm

A previously described differentiation protocol was used [15]. iPSC differentiated into the mesoderm, intermediate mesoderm, Müllerian duct (MD), and nephron progenitors after two, four, and eight days of differentiation, respectively. Organoid formation was achieved 8–30 days after plating on the Matrigel. The FTE-iPSC and NHEK-iPSC morphologies after two and four days exhibited scattered cells in colonies (Figure 3A,B). qPCR revealed pluripotency-(*OCT4*, *SOX2*, and *NANOG*), mesoderm-(*BRACHYURY* and *MIXL1*), and intermediate mesoderm-related genes (*OSR1*, *PAX2*, and *GATA3*) (Figure 3C) on days 2, 4, and 6.

The pluripotency marker *SOX2* was downregulated on day 2. In contrast, the other two markers, *OCT4* and *NANOG*, did not show decreased expression following differentiation in FTEC-iPSCs (Figure 3C, left panel). In NHEK-iPSCs, the expression levels of all three pluripotency markers decreased after differentiation (Figure 3C, right panel).

In Figure 3D, the left panel represents FTEC-iPSC and the right panel represents NHEK-iPSC. There were five culture conditions on days 0, 2, 4, and 6 (WNT3 or WNT4). The mesoderm-related genes (*Brachyury* and *MIXL1*) should be expressed on day 2 of differentiation. The results demonstrated an increased expression of *Brachyury* and *MIXL1* on day 2, which means both iPSCs could differentiate into the mesoderm on day 2. The mesoderm gene expressions (*Brachyury* and *MIXL*) seemed higher in NHEK-iPSC than in FTEC-iPSC with mesoderm differentiation.

The intermediate mesoderm-related genes, *OSR1*, *PAX2*, and *GATA3*, increased in expression on days 4 and 6 of differentiation. In Figure 3E, there was increased expression of OSR1, PAX2, and GATA3 on days 4 and 6, which means both iPSCs were differentiated into the mesoderm on days 4 and 6. The intermediate mesoderm gene expressions, *OSR1* and *PAX2*, were higher in FTEC-iPSC than in the NHEK-iPSC differentiation. The *GATA3* expression was comparable between FTEC-iPSC and NHEK-iPSC.

Taken together, FTEC-iPSC could differentiate toward the mesoderm on day 2 and intermediate mesoderm on days 4 and 6. However, pluripotency genes *OCT4* and *NANOG* were still expressed after differentiation. Compared with FTEC-iPSC, NHEK-iPSC could differentiate into the mesoderm and intermediate mesoderm with decreasing pluripotency gene expressions.

### 3.3. Sequential Differentiation of Both iPSC Lines from Intermediate Mesoderm to FTEC

After intermediate mesoderm differentiation, WNT3A or WNT4 were added to induce differentiation into the FTECs. The differentiated FTEC morphology on days 6 and 8 in both cell lines is shown in Figure 4A,B.

In Figure 4C, the upper panel represents FTEC-iPSC and the lower panel represents NHEK-iPSC. There were five culture conditions on days 0, 2, 4, and 6 (WNT3 or WNT4). The Mullerian duct progenitor-related genes (*WT1* and *OVGP1*) were expressed on day 8 of differentiation. The results demonstrated an increased expression of *WT1* and *OVGP1* on day 8, which means both iPSCs could differentiated into Mullerian duct progenitors by adding WNT3 on day 8. The Mullerian duct progenitor gene expressions (WT1) seemed higher in FTEC-iPSC than in NHEK-iPSC with Mullerian duct progenitor differentiation. The OVGP1 expression was comparable between the two iPSCs.

The nephron progenitor-related genes, *SIX2* and *FOXD1*, showed an increased expression on day 8 of differentiation. In Figure 4D, there was increased expression of *SIX2* or *FOXD1* on days 8 with adding WNT4, which means both iPSCs differentiated into nephron progenitors on days 8. The nephron progenitor gene expression, *SIX2*, was higher in FTEC-iPSC than in NHEK-iPSC differentiation. Conversely, the *FOXD1* expression was higher in NHEK-iPSC than in FTEC-iPSC.

Taken together, FTEC-iPSC could differentiate toward the Mullerian duct progenitor on day 8 after adding WNT3 and the nephron progenitor on day 8 after adding WNT4.

### 3.4. The FTE Organoid Generation from FTE-iPSC-FTE and NHEK-iPSC-FTE

Organoids were formed by plating the FTECs on Matrigel. We used three methods to generate organoids, namely: cells on the Matrigel, spheroids on the Matrigel, and organoids formed in low-attachment dishes.

The organoids formed under all three conditions, and their morphologies are shown in Figure 5A,B. The WNT3 group exhibited a larger and more organized organoid morphology than the WNT4 group.

The organoid in Figure 5A left panel (+WNT3) was a spherical organoid under a microscope. The organoid comprised densely packed cells with a lumen (cavity) in the center. The cells were arranged in various shapes and sizes, but they all had a similar appearance, with a round nucleus and a granular cytoplasm. A layer of columnar epithelial cells surrounded the lumen. These cells had a long, thin shape and were responsible for lining the lumen and absorbing nutrients (Figure 5A). The organoid morphology in Figure 5A right panel (+WNT4) seemed like the organoid in the left panel.

The organoid in Figure 5B left panel (+WNT3) was a tubular organoid under a microscope. The organoid comprised a single layer of cells arranged in a tubular shape. The cells had a cuboidal or columnar shape, a round nucleus, and a granular cytoplasm (Figure 5B). The lumen of the tubule was filled with a fluid called the extracellular matrix (ECM). ECM provided support for the cells and helped to create a microenvironment that was similar to the environment in the body. After adding WNT4, the organoid morphology became densely packed cells without a lumen in the center.

The two organoids differ in their shape and structure. The first organoid was spherical, while the second organoid was tubular. The first organoid had a lumen in the center, while the second organoid had a lumen that ran the tube length. The cells in the two organoids were also different. The cells in the first organoid were arranged in various shapes and sizes, while the cells in the second organoid were arranged in a single layer. Overall, the two organoid images showed two different types of organoids with different shapes and structures. Both organoids could be regarded as FTEC organoids.

IHC also showed that both cell lines formed organoids that expressed the typical FTE markers, namely PAX8, FOXJ1, and LGR5 (Figure 5C). The FTE-related gene expression of PAX8, FOXJ1, and LGR5 increased following organoid differentiation (Figure 5D).

### 3.5. Increased FTE-Related Gene Expression in FTE-iPSC-FTE and NHEK-iPSC-FTE

Finally, we compared the FTE-related gene expression between FTE-iPSC-FTE and NHEK-iPSC-FTE. The *PAX8*, *FOXJ1*, and *LGR5* expression was highest in dish-cultured FTE-iPSC-FTE after adding WNT3 (Figure 6). *PAX8*, *FOXJ1*, and *LGR5* were also increased in dish-cultured NHEK-iPS-FTE after adding WNT3 or WNT4 (Figure 6). Taken together, after adding WNT3, FTE-iPSC-FTE expressed the highest FTE markers compared with the other cell line or culture conditions. However, after adding WNT3 or WNT4, the dish-cultured NHEK-iPS-FTE increased three gene expressions.

## 4. Discussion

We successfully established FTEC-iPSCs, and their morphology reflected the characteristics of stem cell colonies. IHC and qPCR were performed to confirm the expression of pluripotency-related genes. In vitro and in vivo differentiation capabilities were also demonstrated by positive staining for the three germ layer markers and related gene expressions. A sequential differentiation protocol was used to differentiate iPSCs into the mesoderm, intermediate mesoderm, and FTEC organoids. During differentiation, qPCR analysis revealed the presence of pluripotent, mesodermal, and intermediate mesodermal-related genes. Following intermediate mesodermal differentiation, WNT3A addition drove the cells toward an FTEC morphology, as confirmed by the FTEC-related marker expression and tubule-like structure formation. The Mullerian duct progenitor-related genes were increased earlier, since day 4, in the differentiated iPSC from FTEC than from NHEK. The FTEC-related gene expression was increased in FTECs differentiated from FTEC-iPSCs and those from NHEK-iPSCs in a dish-culture condition, plus WNT3.

The cell type origin may influence the functional and molecular characteristics of iPSCs [19]. A previous study reported that the cell type origin could affect the differentiation abilities of iPSCs [19]. However, the epigenetic memory of these original cell types is attenuated after an extended passage period [19]. Another study showed that the disease status of the cell origin may influence the differentiation potential [22]. They further found that iPSCs derived from healthy chondrocytes had better chondrocyte differentiation abilities than those derived from osteoarthritis chondrocytes [22]. Furthermore, another study showed that the differentiation ability of iPSCs derived from cardiomyocytes is better than that of iPSCs derived from skin fibroblasts [23]. Our analysis agrees with these studies, showing that iPSCs derived from FTECs have better FTEC differentiation capabilities.

WNT signaling is essential for FTEC differentiation [24]. A previous study reported adding WNT4 to induce MD progenitors and FTEC differentiation [15]. However, in our study, we found that WNT3A addition resulted in improved MD progenitor and FTEC differentiation compared with the addition of WNT4. In our previous studies, WNT3A was established as an essential factor for FTECs or FTEC organoid culturing [13,25]. Therefore, WNT3A may play an essential role in FTEC differentiation induction.

In our study, the gene expression levels of FTEC markers (*PAX8*, *FOXJ1*, and *LGR5)* in the organoids were comparable to the original FTEC cells. This is the best reference available and organoids with similar expression levels should be considered for further research [15].

The long-term stability of our model is critical for the pharmacological studies of potential therapy. Proper culture techniques and maintenance protocols are essential to ensure the long-term stability of organoids [26]. Cryopreservation techniques can be used to store organoids long term, providing a valuable resource for future experiments [27]. Stable organoids are reliable platforms for high-throughput drug screening. They can help identify promising drug candidates and assess their efficacy and safety [28]. Improving the long-term stability of organoids by refining culture methods ensures that organoid models remain relevant and reliable for a wide range of applications, including drug development and therapy testing [29].

The main strength of this study lies in the fact that we compared FTEC-iPSCs to primary FTEC in terms of gene expression. Two cell types (FTEC and NHEK) were used to generate iPSCs, and their differentiation capabilities were compared. However, this study has several limitations: (1) We used only two cell types (FTEC and NHEK) for iPSC generation. Therefore, these findings may not apply to iPSCs derived from other cell types. (2) The study did not investigate the long-term stability of the FTEC-iPSC differentiation capability. This study evaluated only the ability of FTEC-iPSCs to differentiate into FTEC. (3) The potential of the iPSCs established in this study for other applications, such as disease modeling and drug screening, has not been explored. (4) Finally, this study did not investigate the molecular mechanisms underlying the improved differentiation ability of FTEC-iPSCs compared with NHEK-iPSCs. This study only evaluated the effect of WNT3A addition on FTEC differentiation and did not investigate other factors that may have also influenced this process. We found two different types of organoids with different shapes and structures. A mixture of different types of organoids makes it more difficult to obtain robust and reproducible results. Overall, although this study demonstrates the potential of FTEC-iPSCs for generating FTEC organoids, further studies are required to understand the limitations and potential applications of this approach fully.

## 5. Conclusions

Our study successfully generated iPSCs from FTECs, demonstrating their capacity for FTEC differentiation. Furthermore, iPSCs originating from orthologous cell sources exhibited comparable differentiation capabilities. These findings hold promise for using iPSCs in modeling and investigating diseases associated with these specific cell types.

## Figures and Tables

**Figure 1 cells-12-02635-f001:**
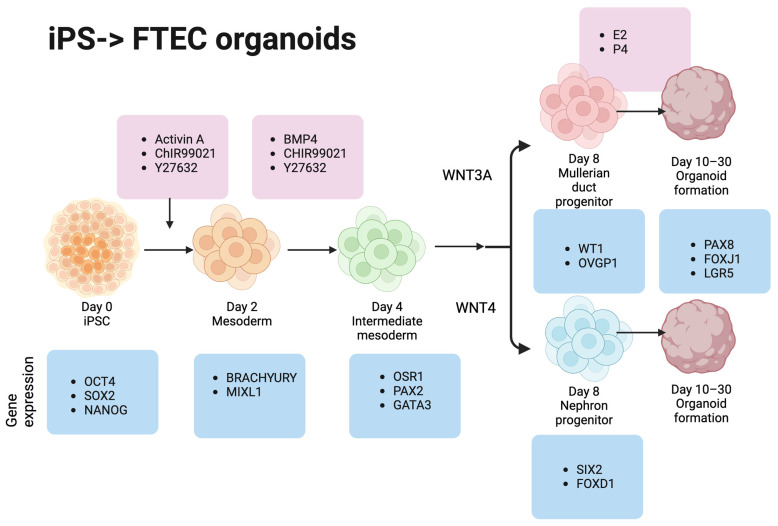
The experimental design from induced pluripotent stem cells (iPSCs) to fallopian tube epithelial cells (FTECs).

**Figure 2 cells-12-02635-f002:**
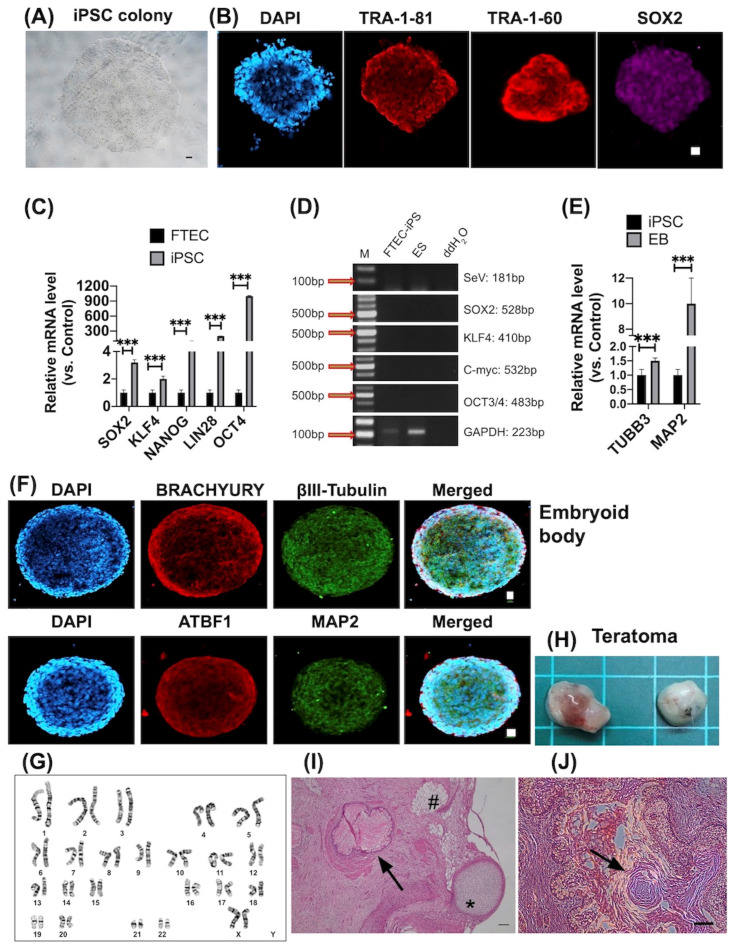
Characterization of fallopian tube epithelial cell (FTEC)-induced pluripotent stem cells (iPSCs). (**A**) Morphology of colony formed by FTEC-iPSCs. Scale bar = 100 μm. (**B**) Immunohistochemistry (IHC) reveals pluripotency protein expressions, including SOX2, TRA-1-60, and TRA-1-81. DAPI indicates nuclear staining. SOX2 is nuclear staining, and TRA-1-60 and TRA-1-81 are cytoplasmic staining. Scale bar = 20 μm. (**C**) Quantitative PCR shows pluripotency gene expressions, including *OCT4*, *SOX2*, *NANOG*, *KLF4*, and *LIN28* in FTEC-iPSC. *** *p* < 0.001. (**D**) RT-PCR shows no Sendai-virus-related pluripotency gene expressions in passage 6 of FTEC-iPSC. ES: embryonic stem cells. ddH2O: double-distilled water. (**E**) qPCR reveals ectoderm differentiation of embryoid bodies. *TUBB3* and *MAP2* have increased expression after differentiation. *** *p* < 0.001. (**F**) IHC reveals embryoid bodies’ differentiation of FTE-iPSCs, including BRACHYURY (mesoderm), βIII-tubulin and MAP2 (ectoderm), and ATBF1 (endoderm). (**G**) Chromosome analysis of FTEC-iPSCs (passage 13). It shows normal karyotype 46, XX. The number in the figure is referred to the chromosome number. (**H**) iPSCs form a teratoma in mice (*n* = 2). Scale = 1 cm. (**I**) iPSC forms a teratoma histology showing three germ layer tissues. * Cartilage, # fat tissue (mesoderm), arrow: endoderm tissue, (**J**) arrow: ectoderm. Scale bar = 100 μm.

**Figure 3 cells-12-02635-f003:**
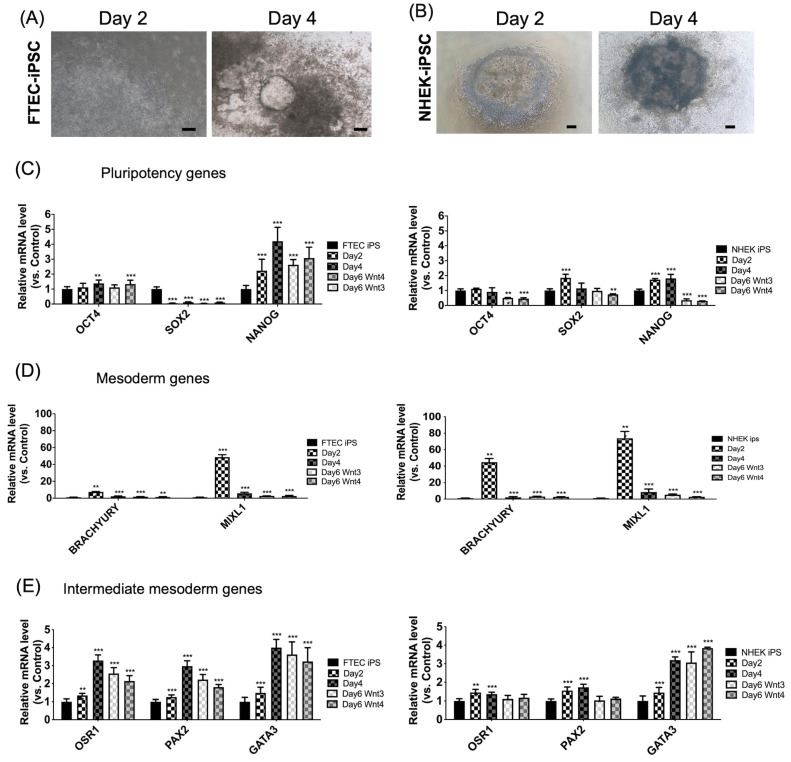
Morphology and gene expressions of pluripotency, mesoderm, and intermediate mesoderm-related genes in induced pluripotent stem cells (iPSC) derived from fallopian tube epithelial cells (FTEC) and normal human keratinocyte (NHEK) on days 0, 2, 4, and 6. (**A**,**B**) The morphology of differentiated FTEC from FTEC-iPSC (**A**) and NHEK-iPSCs (**B**) on days 2 and 4. (**C**–**E**) Gene expression of pluripotency (**C**), mesoderm (**D**), and intermediate mesoderm (**E**)-related genes of FTEC-iPSCs and NHEK-iPSCs on days 0, 2, 4, and 6. ** *p* < 0.01, *** *p* < 0.001. Scale bar = 100 μm. Pluripotency genes gradually decrease after differentiation in iPSC derived from NHEK. In both cell lines, mesoderm and intermediate mesoderm-related genes increase after differentiation on day 2 and day 4, respectively.

**Figure 4 cells-12-02635-f004:**
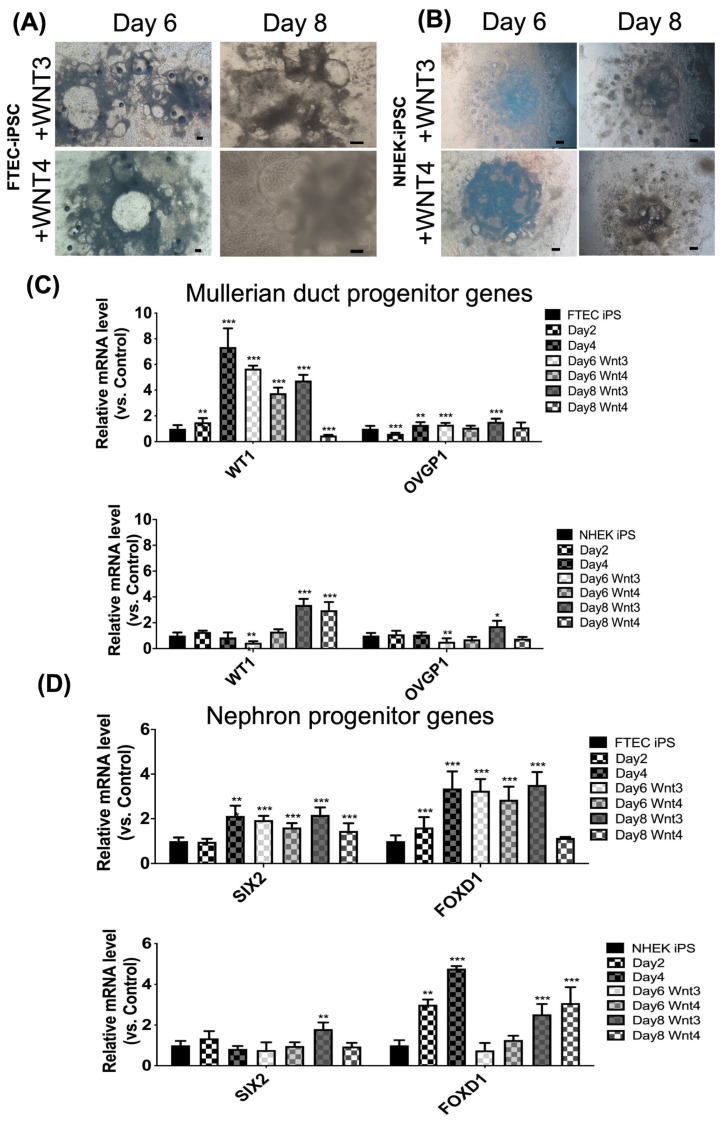
Morphology and gene expressions of Mullerian duct progenitor- and nephron progenitor-like cell differentiation from iPSCs derived from fallopian tube epithelial cells (FTECs) and normal human keratinocytes (NHEK) on days 6 and 8. (**A**,**B**) The morphology of differentiated FTEC from FTEC-iPSC (**A**) and NHEK-iPSC (**B**) on days 6 and 8 after adding WNT3 or WNT4. Scale bar = 100 μm. (**C**,**D**) Gene expression of Mullerian duct progenitor (*WT1* and *OVGP1*) (**C**) and nephron progenitor-related genes (*SIX2* and *FOXD1*) (**D**) of differentiated FTEC from FTEC-iPSCs and NHEK-iPSCs on days 0, 2, 4, 6, and 8 (*n* = 3). * *p* < 0.05, ** *p* < 0.01, *** *p* < 0.001. The Mullerian duct progenitor-related genes are increased in differentiated iPSC on day 8 plus WNT3, but not in plus WNT4. Nephron progenitor-related genes are also increased in differentiated iPSCs from FTEC on day 8 plus WNT3 but not in plus WNT4. Nephron progenitor-related genes are also increased in differentiated iPSCs from NHEK on day 8 plus WNT3 or WNT4. The Mullerian duct progenitor-related genes are increased earlier, since day 4 in differentiated iPSC from FTEC than from NHEK.

**Figure 5 cells-12-02635-f005:**
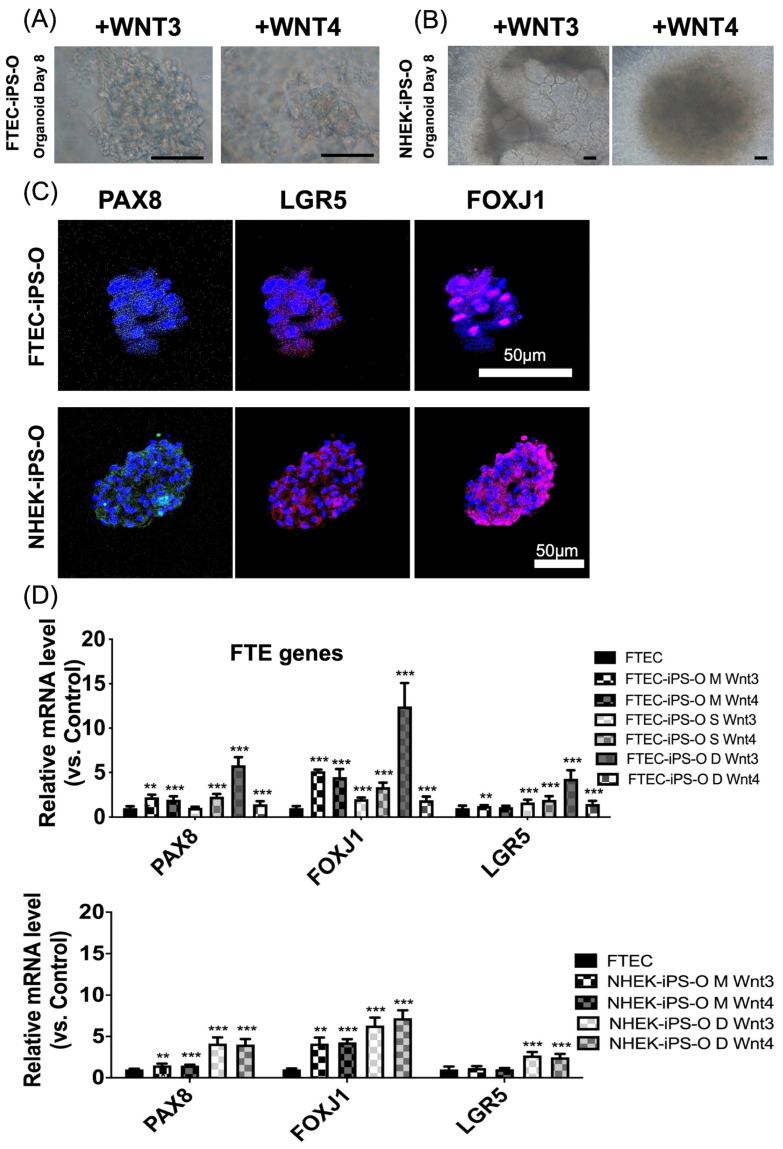
Morphology, protein, and gene expressions of organoids differentiated from induced pluripotent stem cells (iPSCs) of fallopian tube epithelial cells (FTEC) and normal human keratinocytes (NHEK). (**A**,**B**) The morphology of organoids after 8 days of organoids differentiation from iPSCs from FTEC (**A**) and NHEK (**B**). Spherical organoid (**A**) and tubular organoid (**B**) are noted. (**C**) Both cell lines could form organoids. Immunohistochemistry (IHC) showed positive staining of PAX8 (green), LGR5 (red), and FOXJ1 (pink) in the organoids differentiated from iPSC derived from FTEC and NHEK. DAPI (blue) is used to stain the nucleus. (**D**) qPCR shows increasing *PAX8* (secretory cell marker), *FOXJ1* (ciliated cell marker), and *LGR5* (stem cell marker) expression in the organoids derived from the iPSC derived from FTEC and NHEK in three culture conditions plus WNT3 or WNT4 (*n* = 3). ** *p* < 0.01, *** *p* < 0.001. Scale bar = 100 μm. M: Matrigel, D: low attach dish. S: spheroid. O: organoid.

**Figure 6 cells-12-02635-f006:**
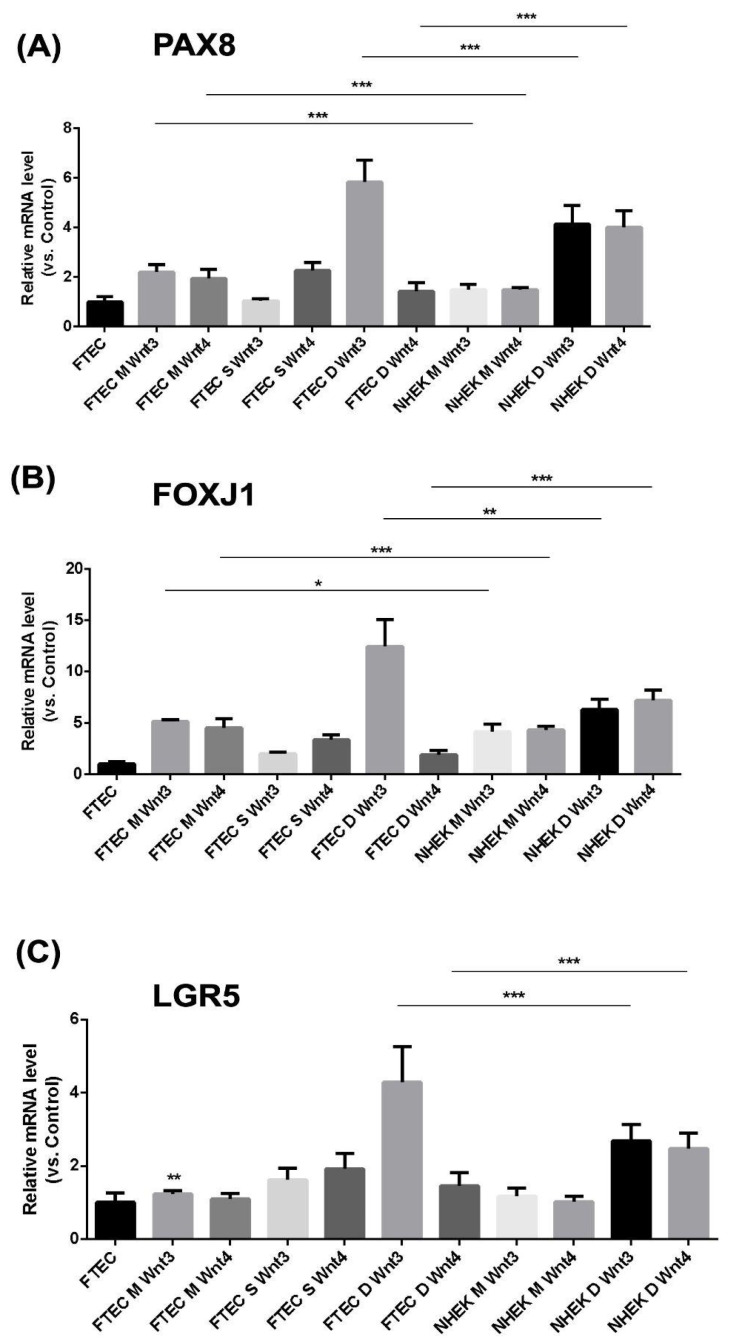
Comparison of FTEC-related gene expressions between the organoids derived from iPSCS from fallopian tube epithelial cells (FTECs) and normal human keratinocytes (NHEK). (**A**) *PAX8*, (**B**) *FOXJ1*, and (**C**) *LGR5* (*n* = 3). The organoids derived from iPSC originating from FTEC in the dish culture method with the addition of WNT3 exhibit the highest expression for the three genes. The second-highest expression of these three genes is observed in the organoids derived from iPSC originating from NHEK with the supplementation of both WNT3 and WNT4. * *p* < 0.05, ** *p* < 0.01, *** *p* < 0.001. M: Matrigel, D: low attach dish. S: spheroid.

**Table 1 cells-12-02635-t001:** The primer sequence of the tested genes.

Gene Name	Forward Sequence	Reverse Sequence	Product Size (bp)
*Oct4*	CAG TGC CCG AAA CCC ACA C	CAG TGC CCG AAA CCC ACA C	161
*Nanog*	AGT CCC AAA GGC AAA CAA CCC ACT TC	TGC TGG AGG CTG AGG TAT TTC TGT CTC	161
*Sox2*	GGG AAA TGG GAG GGG TGC AAA AGA GG	TTG CGT GAG TGT GGA TGG GAT TGG TG	151
*Lin28*	TGCACCAGAGTAAGCTGCAC	CTCCTTTTGATCTGCGCTTC	189
*Klf4*	CCCACACAGGTGAGAAACCT	ATGTGTAAGGCGAGGTGGTC	169
*MAP2*	GCA TGA GCT CTT GGC AGG	CCA ATT GAA CCC ATG TAA AGC C	194
*GFAP*	AGG GCT GAC ACG TCC AC	GCC TTA GAG GGG AGA GGA G	132
*GATA4*	TCC CTC TTC CCT CCT CAA AT	TCA GCG TGT AAA GGC ATC TG	194
*Tubb3*	CAG AGC AAG AAC AGC AGC TAC TT	GTG AAC TCC ATC TCG TCC ATG CCC TC	227
*Hand1*	TGC CTG AGA AAG AGA ACC AG	ATG GCA GGA TGA ACA AAC AC	274
*GATA6*	CCT CAC TCC ACT CGT GTC TGC	GTC CTG GCT TCT GGA AGT GG	225
*MIXL1*	GGTACCCCGACATCCACTT	GCCTGTTCTGGAACCATACCT	87
*BRACHYURY*	GCTGTGACAGGTACCCAACC	CATGCAGGTGAGTTGTCAGAA	106
*PAX2*	GAAGTGCCCCCTTGTGTG	TCGTTGTAGGCCGTGTACTG	82
*OSR1*	GGACCTCTGCGGAACAAG	TGCAGGGAAGGGTGGATA	100
*GATA3*	CTCATTAAGCCCAAGCGAAG	GTCTGACAGTTCGCACAGGA	68
*SIX2*	CAGGTCAGCAACTGGTTCAA	AGCTGCCTAACACCGACTTG	136
*FOXD1*	GACTCTGCACCAAGGGACTG	CAATTGGAAATCCTAGCAGTAAAGT	63
*FOXJ1*	GGGGTGGGAGCAACTTCT	CCTCCTCCGAATAAGTATGTGGT	83
*PAX8*	CAACAGCACCCTGGACGAC	AGGGTGAGTGAGGATCTGCC	113
*WT1*	GAATGCATGACCTGGAATCA	TCTGCCCTTCTGTCCATTTC	94
*OVGP1*	AAGCTGTTGCTGTGGGTTG	TGTGCCCAGTTGGTGAAAT	92
*GAPDH*	GGTCTCCTCTGACTTGAACA	GTGAGGGTCTCTCTCTTCCT	221
RT-PCR primer			
*SeV*	GGA TCA CTA GGT GAT ATC GAG C	ACC AGA CAA GAG TTT AAG AGA TAT GTA TC	181
*SOX2*	ATG CAC CGC TAC GAC GTG AGC GC	ACC TTG ACA ATC CTG ATG TGG	528
*KLF4*	TTC CTG CAT GCC AGA GGA GCC C	AAT GTA TCG AAG GTG CTC AA	410
*C-MYC*	TAA CTG ACT AGC AGG CTT GTC G	TCC ACA TAC AGT CCT GGA TGA TGA TG	532
*OCT3/4*	CCC GAA AGA GAA AGC GAA CCA G	AAT GTA TCG AAG GTG CTC AA	483
*GAPDH*	CCA TCT TCC AGG AGC GAG	GCA GGA GGC ATT GCT GAT	233

RT-PCR: reverse transcription-polymerase chain reaction.

## Data Availability

The data supporting the conclusions of this article is included within the article.

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
