# Peer review of "Assessment of Fallopian Tube Epithelium Features Derived from Induced Pluripotent Stem Cells of Both Fallopian Tube and Skin Origins"

_cells, 2023, doi:10.3390/cells12222635_

Round 1

Reviewer 1 Report

Comments and Suggestions for Authors

In this study the authors develop a new iPSC line starting from fallopian tube epithelial cells and compare their differentiation capability with skin keratinocyte-derived iPSCs. It is an interesting work that may be useful to study the pathophysiology of the ovarian cancer. However, the long-term stability of this cell line differentiation was not investigated. This is an important aspect of the research, especially for future potential use of this model in pharmacological studies. The authors fully reported this critical aspect in the discussion.

However, I have some questions:

All the figures show gene expression levels in iPSC derived from fallopian tube epithelial cell and normal human keratinocyte. What about protein expression? Immunohistochemistry was used only to analyze pluripotency protein expressions in Fig.2 and to evaluate differentiation in organoids derived iPSCs of fallopian tube epithelial cells and normal human keratinocytes in Fig.5.

Have you performed also western blot?

Is it comparable with gene expression?

Author Response

Responses to Reviewer 1’s comments

We would like to thank the reviewer for the valuable and constructive comments. We have taken all the remarks into account and have revised the manuscript accordingly. The revised portions are highlighted in red font.

Comment 1: It is an interesting work that may be useful to study the pathophysiology of the ovarian cancer.

Response 1: We thank the reviewer for the positive feedback.

Comment 2: However, the long-term stability of this cell line differentiation was not investigated. This is an important aspect of the research, especially for future potential use of this model in pharmacological studies. The authors fully reported this critical aspect in the discussion.

Response 2: We thank the reviewer for this excellent suggestion. We fully agree with the reviewer regarding the viewpoint the long-term stability of our model is critical for the pharmacological studies of potential therapy. The current study mainly focused on the establishment of this differentiation model. In fact, we have planned the test of the long-term stability of our model alongside pharmacological interventions in the follow-up study. The suggestion from the reviewer substantially enhances the credibility of our follow-up study.

In response to the reviewer’s suggestion, we have fully reported this critical aspect in a new paragraph of the discussion section (page 18, lines 448-456). The paragraph read as: “The long-term stability of our model is critical for the pharmacological studies of potential therapy. Proper culture techniques and maintenance protocols are essential to ensure the long-term stability of organoids [27]. Cryopreservation techniques can be used to store organoids long-term, providing a valuable resource for future experiments [28]. Stable organoids are reliable platforms for high-throughput drug screening. They can help identify promising drug candidates and assess their efficacy and safety [29].  Improving the long-term stability of organoids by refining culture methods ensures that organoid models remain relevant and reliable for a wide range of applications, including drug development and therapy testing [30].”

Comment 3: All the figures show gene expression levels in iPSC derived from fallopian tube epithelial cell and normal human keratinocyte. What about protein expression? Immunohistochemistry was used only to analyze pluripotency protein expressions in Fig.2 and to evaluate differentiation in organoids derived iPSCs of fallopian tube epithelial cells and normal human keratinocytes in Fig.5.

Response 3: We thank the reviewer for the suggestion. Unfortunately, the amount of differentiation cells was not sufficient to perform the Western blot analysis of protein expression.

We did, however, add qPCR data regarding TUBB3 and MAP2 as Figure 2E to provide additional data to support the gene expression profile shown in Figure 2E (page 9). We sincerely hope the reviewer can understand our obstacle for not being able to perform Western blot analysis.

Comment 4: Have you performed also western blot? Is it comparable with gene expression?

Response 4:  Unfortunately, the amount of differentiation cells was not sufficient to perform the Western blot analysis of protein expression. We sincerely hope the reviewer can understand our obstacle for not being able to perform Western blot analysis.

Reviewer 2 Report

Comments and Suggestions for Authors

In this article, the authors describe the generation of iPSCs reprogramed from FTECs, a cell type involved in ovarion cancer pathology. The iPSCs were characterized for their pluripotent potential. Moreover, iPSCs derived from FTEC and previously generated iPSCs derived from skin keratinocytes (NHEK) were differentiated into FTECs using published protocols. The authors further evaluated the differentiation efficiency comparing these two lines.

While the research question is interesting and the authors provide new data regarding the reprogramming of FTECs and the differentiation potential of the newly created iPSCs, the conclusion remains unclear to me. 

Major points

  • Figure 3. What exactly is the result of differentiating these two iPSC lines into mesodermal precursor cells? Are there differences between the differentiation experiments? At least the scale bars should be made comparable between B and D.
  • Figure 4. Same as for figure 3.
  • Figure 5 (A+D). The structures shown are quite blurry. Comparing FTEC and NHEK differentiations it looks like as if there are completely different cellular and structural morphologies. Can the authors comment on that? What is to be expected at this stage of differentiation?
  • Figure 5 (B+E). Fluorescence images must show the same markers for FTEC and NHEK differentiations. 
  • Figure 5 (C+F). Again, the scale bars should be adjusted so that a comparison between values in C and F is possible.
  • The authors write that “FTE-iPSC-FTE could express highest FTE markers” (page 12, line 375) and that “The 397 FTEC-related gene expression was higher in FTECs differentiated from FTEC-iPSCs than those from NHEK-iPSCs in a dish-culture condition plus adding WNT3” (page 13, lines 397-399). However, is the highest expression always the desired result? Wouldn’t it be best to reach the same expression level as assessed for the original FTECs (not higher and not lower)?
  • Can the authors include additional analysis to characterize the organoids and compare it to the original FTECs, such as western blots probed with markers used for the immunostainings? As these images have not been analyzed quantitively, there needs to be an additional method, other than gene expression levels of three genes.

Minor points

  • Page 2, line 96. What is meant with “we developed a new technique for iPSC derivation”? As far as I understood, the authors used a kit to generate iPSCs.
Comments on the Quality of English Language

Moderate editing of English language required.

Author Response

Responses to Reviewer 2’s comments

We would like to thank the reviewer for the valuable and constructive comments. We have taken all the remarks into account and have revised the manuscript accordingly. The revised portions are highlighted in red font.

Comment 1: While the research question is interesting and the authors provide new data regarding the reprogramming of FTECs and the differentiation potential of the newly created iPSCs, the conclusion remains unclear to me.

Response 1: We thank the reviewer for the suggestion. In response to the suggestion, the conclusion has been written for clarity. The statements read as:” Our study successfully generated iPSCs from FTECs, demonstrating their capacity for FTEC differentiation. Furthermore, iPSCs originating from orthologous cell sources exhibited comparable differentiation capabilities. These findings hold promise for using iPSCs in modeling and investigating diseases associated with these specific cell types. (page 19, lines 475-478)”

Major points

Comment 2: Figure 3. What exactly is the result of differentiating these two iPSC lines into mesodermal precursor cells? Are there differences between the differentiation experiments? At least the scale bars should be made comparable between B and D.

Response 2: We thank the reviewer’s suggestion.

Regarding the first question, in Figure 3D, left panel represented FTEC-iPSC, right panel represented NHEK-iPSC. There were five culture conditions on days 0, 2, 4, and 6 (WNT3 or WNT4). The mesoderm-related genes (Brachyury and MIXL1) should be expressed on day 2 of differentiation. The results demonstrated increased expression of Brachyury and MIXL1 on day 2, which means both iPSCs could differentiated into mesoderm on day 2. The intermediate mesoderm-related genes, OSR1, PAX2, and GATA3, should increase expression on day 4 of differentiation. In Figure 2E, increased expression of OSR1, PAX2, and GATA3 on days 4 and 6, which means both iPSCs could differentiated into mesoderm on days 4 and 6. (page, 10, lines 296-308)

Regarding the second question, the mesoderm gene expressions (Brachyury and MIXL) seemed higher in NHEK-iPSC than in FTEC-iPSC with mesoderm differentiation. The intermediate mesoderm gene expression, OSR1 and PAX2,  were higher in FTEC-iPSC than in NHEK-iPSC differentiation. The GATA3 expression was comparable between FTEC-iPSC and NHEK-iPSC. (page, 10, lines 296-308)

Regarding the suggestion of scale bars, the panels in Figure 3 have been rearranged and scale bars were comparable between the left and right panels of Figures 3 C, D, and E (page 11).

Comment 3: Figure 4. Same as for figure 3.

Response 3: We thank the reviewer’s suggestion.

Regarding the first question, in Figure 4C, upper panel represented FTEC-iPSC, lower panel represented NHEK-iPSC. There were five culture conditions on days 0, 2, 4, and 6 (WNT3 or WNT4). The Mullerian duct progenitor-related genes (WT1 and OVGP1) should be expressed on day 8 of differentiation. The results demonstrated increased expression of WT1 and OVGP1 on day 8, which means both iPSCs could differentiate into Mullerian duct progenitors with adding WNT3 on day 8. The nephron progenitor-related genes, SIX2 and FOXD1, should increase expression on day 8 of differentiation. In Figure 4D, increased expression of SIX2 or FOXD1 on days 8 with adding WNT4, which means both iPSCs could differentiated into nephron progenitor on days 8.  (page 12, lines 328-344)

Regarding the second question, the Mullerian duct progenitor gene expressions (WT1) seemed higher in FTEC-iPSC than in NHEK-iPSC with Mullerian duct progenitor differentiation. The OVGP1 expression was comparable between the two iPSC. The nephron progenitor gene expression, SIX2 was higher in FTEC-iPSC than in NHEK-iPSC differentiation. Conversely, the FOXD1 expression was higher in NHEK-iPSC than in FTEC-iPSC. (page, 12, lines 328-344)

Regarding the suggestion of scale bars, the panels in Figure 4 have been rearranged and scale bars were comparable between the upper and lower panels of Figures 4 C and D (page 13).

Comment 4: Figure 5 (A+D). The structures shown are quite blurry. Comparing FTEC and NHEK differentiations it looks like as if there are completely different cellular and structural morphologies. Can the authors comment on that? What is to be expected at this stage of differentiation?

Response 4: We thank the reviewer for the suggestion. Please note that the panels in Figure 5 have been rearranged. In response to the suggestion, we have added statements to explain the structure morphologies of differentiation organoids (page 14, lines 365-386). The statements read as: “The organoid in Fig. 5A left panel (+WNT3) was a spherical organoid under a microscope. The organoid comprised densely packed cells with a lumen (cavity) in the center. The cells were arranged in various shapes and sizes, but they all had a similar appearance, with a round nucleus and a granular cytoplasm. A layer of columnar epithelial cells surrounded the lumen. These cells had a long, thin shape and were responsible for lining the lumen and absorbing nutrients (Fig. 5A). The organoid morphology in Fig. 5A right panel (+WNT4) seemed like the organoid in the left panel.

The organoid in Fig. 5B left panel (+WNT3) was a tubular organoid under a microscope. The organoid is comprised of a single layer of cells arranged in a tubular shape. The cells had a cuboidal or columnar shape, a round nucleus, and a granular cytoplasm (Fig 5B). The lumen of the tubule was filled with a fluid called extracellular matrix (ECM). The ECM provided support for the cells and helped to create a microenvironment that was similar to the environment in the body. After adding WNT4, the organoid morphology became densely packed cells without a lumen in the center.

The two organoids differ in their shape and structure. The first organoid was spherical, while the second organoid was tubular. The first organoid had a lumen in the center, while the second organoid had a lumen that ran the tube length. The cells in the two organoids were also different. The cells in the first organoid were arranged in various shapes and sizes, while the cells in the second organoid were arranged in a single layer. Overall, the two organoid images showed two different types of organoids with different shapes and structures. This diversity of organoids was one of the things that made them such a powerful tool for research. Both organoids could be regarded as FTEC organoids.”

Comment 5: Figure 5 (B+E). Fluorescence images must show the same markers for FTEC and NHEK differentiations.

Response 5: We fully agree with the reviewer’s suggestion. In response to this suggestion, we have provided additional fluorescence images of FTEC and NHEK in Figure 5C to show the same types of proteins in Figure 5D (PAX8, FOXJ1, and LGR5).

Comment 6: Figure 5 (C+F). Again, the scale bars should be adjusted so that a comparison between values in C and F is possible.

Response 6: We thank the reviewer’s suggestion. We have redone Figure 5 and made scale bars comparable between C and F (Figure D upper panel and lower panel in new Figure 5).

Comment 7: The authors write that “FTE-iPSC-FTE could express highest FTE markers” (page 12, line 375) and that “The 397 FTEC-related gene expression was higher in FTECs differentiated from FTEC-iPSCs than those from NHEK-iPSCs in a dish-culture condition plus adding WNT3” (page 13, lines 397-399). However, is the highest expression always the desired result? Wouldn’t it be best to reach the same expression level as assessed for the original FTECs (not higher and not lower)?

Response 7: We thank the reviewer for the suggestion. Since these FTECs were derived from iPSCs, we are not sure whether the gene expression should be the same as the original FTECs. We do, however, recognize that the increases in these genes could serve as indicators or markers for the differentiation of FTECs. We sincerely hope that the reviewer can approve our explanation.

Comment 8: Can the authors include additional analysis to characterize the organoids and compare it to the original FTECs, such as western blots probed with markers used for the immunostainings? As these images have not been analyzed quantitively, there needs to be an additional method, other than gene expression levels of three genes.

Response 8: We thank the reviewer for the suggestion. Unfortunately, the amount of differentiation cells in the organoids was not sufficient to perform the Western blot analysis of protein expression. We sincerely hope the reviewer can understand our obstacle for not being able to perform Western blot analysis.

Minor points

Comment 9: Page 2, line 96. What is meant with “we developed a new technique for iPSC derivation”? As far as I understood, the authors used a kit to generate iPSCs.

Response 9: We thank the reviewer for reminding us of this issue. The introduction section has been substantially rewritten. This sentence has been deleted.

Comment 10: Comments on the Quality of English Language

Moderate editing of English language required.

Response 10: In response to the suggestion, the revised manuscript has been carefully proofread by the authors and an external expert.

Reviewer 3 Report

Comments and Suggestions for Authors

The study is interesting and got a good results. However, it should be condisered followings;

1) It is neededd reorganization the abstract and introduction to improve readability. 

2) Needed full information for materials

     e.g. the catalog number for DMEM etc...

3) Improve the quality of figures

   - the legned pont is too small and hard to read.

   - to give information of tissue or organized structure (Fig 3, 4, 5)

Comments on the Quality of English Language

It is needed reorganization of the introduction and abstract. 

Author Response

Responses to Reviewer 3’s comments

We would like to thank the reviewer for the valuable and constructive comments. We have taken all the remarks into account and have revised the manuscript accordingly. The revised portions are highlighted in red font.

Comment 1: It is neededd reorganization the abstract and introduction to improve readability.

Response 1: We thank the reviewer for the suggestion. In response to this suggestion, the abstract and introduction sections have been substantially written and reorganized to improve readability (pages 1 and 2).

Comment 2: Needed full information for materials. e.g. the catalog number for DMEM etc…

Response 2: We thank the reviewer for careful reading of our manuscript. In response to the suggestion, detailed information for materials including the catalog number of the products has been provided in the methods section.

Comment 3: Improve the quality of figures

   - the legned pont is too small and hard to read.

   - to give information of tissue or organized structure (Fig 3, 4, 5)

Response 3: We thank the reviewer for the suggestion. In response to the suggestion, the sizes of the figures and the legend font have been enlarged. Also, the information on the tissues used in each panel in Figures 3, 4, and 5 has been added.

Comment 4: Comments on the Quality of English Language

It is needed reorganization of the introduction and abstract.

Response 4: We thank the reviewer for the suggestion. In response to this suggestion, the abstract and introduction sections have been substantially written and reorganized (pages 1 and 2).  

Round 2

Reviewer 2 Report

Comments and Suggestions for Authors

The revised manuscript has been improved significantly. I only have a few remaining minor comments.

·     The authors should add the information on the organoid type (spherical or tubular) in the figure legend of Figure 5. 

·     The authors replied (Response 4) “Overall, the two organoid images showed two different types of organoids with different shapes and structures. This diversity of organoids was one of the things that made them such a powerful tool for research. Both organoids could be regarded as FTEC organoids.“ In my view, the diversity of organoids is rather a challenge and not an advantage. A mixture of different types of organoids makes it more difficult to obtain robust and reproducible results. This should be discussed as a limitation of the technique.

·     The authors replied (Response 7) “Since these FTECs were derived from iPSCs, we are not sure whether the gene expression should be the same as the original FTECs.” I suggest adding a statement in the Discussion section that gene expression levels of FTEC markers in organoids were comparable to original FTEC cells. This is the best reference available and organoids with similar expression levels are to be considered for further research.

Comments on the Quality of English Language

Minor editing of English language required

Author Response

Response to Reviewer 2

The revised manuscript has been improved significantly. I only have a few remaining minor comments.

Comment 1. The authors should add the information on the organoid type (spherical or tubular) in the figure legend of Figure 5. 

Response 1: We thank the reviewer for the suggestion. In response to the suggestion, the information on the organoid type has been added in the legend of Figure 5. The statements read as: ”Spherical organoid (A) and tubular organoid (B) were noted.” (page. 16, lines 393-394)

Comment 2. The authors replied (Response 4) “Overall, the two organoid images showed two different types of organoids with different shapes and structures. This diversity of organoids was one of the things that made them such a powerful tool for research. Both organoids could be regarded as FTEC organoids.“ In my view, the diversity of organoids is rather a challenge and not an advantage. A mixture of different types of organoids makes it more difficult to obtain robust and reproducible results. This should be discussed as a limitation of the technique.

Response 2: We fully agree with the viewpoint of the reviewer. In response to the suggestion, we have deleted the statement regarding the advantage: “This diversity of organoids was one of the things that made them such a powerful tool for research.” Also, we have added statements in the limitation section regarding the challenge of the technique. These statements read as: “We found two different types of organoids with different shapes and structures. A mixture of different types of organoids makes it more difficult to obtain robust and reproducible results.” (page 19, lines 472-474)

Comment 3. The authors replied (Response 7) “Since these FTECs were derived from iPSCs, we are not sure whether the gene expression should be the same as the original FTECs.” I suggest adding a statement in the Discussion section that gene expression levels of FTEC markers in organoids were comparable to original FTEC cells. This is the best reference available and organoids with similar expression levels are to be considered for further research.

Response 3: We thank the reviewer for this excellent suggestion. In response to the suggestion, we have added statements to the discussion section. These statements read as: ” In our study, the gene expression levels of FTEC markers (PAX8, FOXJ1, and LGR5) in organoids were comparable to original FTEC cells. This is the best reference available and organoids with similar expression levels are to be considered for further research [15].” (page 18, lines 448-450)

Comments on the Quality of English Language

Minor editing of English language required

Response 4: In response to the suggestion, the revised manuscript has been carefully proofread by the authors and an external expert.